# BEACON: THWARTING BACKDOOR ATTACKS IN CROSS-DOMAIN FEDERATED FINE-TUNING VIA GRADIENT BEHAVIOR DECOUPLING

## ABSTRACT

Cross-domain federated fine-tuning (CD-FFT) has emerged as a promising paradigm evolving from traditional federated learning (FL), with better alignment to real-world data distributions and enhanced communication efficiency. However, the inherent domain shift and rapid local adaptation in CD-FFT substantially amplify its susceptibility to backdoor attacks. Existing studies have just revealed the vulnerability of CD-FFT to backdoor threats, but fall short of exploring robust defense solutions. To bridge this gap, we first systematically evaluate the transferability of existing FL backdoor defenses to the CD-FFT setting, revealing their limited effectiveness under this more challenging scenario. Motivated by this, we propose BEACON, an innovative backdoor defense framework that decouples gradient behaviors at a fine granularity to uncovers malicious signals. Specifically, we creates a novel *Task-Deviation Orthogonal Disentanglement* (TDOD) module, which orthogonally decomposes client updates into consensus and deviation components, enabling joint reasoning over benign contribution and suspicious divergence. Furthermore, a *Classification Head Inconsistency Forensics* module is designed to capture boundary-shifting artifacts by traversing per-class gradients, thus identifying label-wise anomalies indicative of targeted tampering. Consequently, BEACON enables effective, robust, and domain-adaptive backdoor defense in CD-FFT. Extensive experiments across four cross-domain benchmarks and three backdoor variants demonstrate that BEACON consistently suppresses attack success rates to below 2%, while preserving main task accuracy, significantly outperforming seven state-of-the-art defenses in this challenging setting.

## 1 INTRODUCTION

Federated learning (FL) (McMahan et al., 2017; Kairouz et al., 2021; Ye et al., 2023) is a distributed learning paradigm designed to preserve data privacy, and has been widely adopted in various real-world applications such as healthcare (Nguyen et al., 2022a), remote sensing (Moreno-Alvarez et al., 2024), and personalized recommendation systems (Feng et al., 2024). However, as models scale up and data distributions become increasingly complex, FL has progressively shifted toward settings involving domain discrepancies across clients (Li et al., 2020; Zhang et al., 2023b; Chen et al., 2023), while also has embracing parameter-efficient fine-tuning strategies (Hu et al., 2021; Jia et al., 2022; Lian et al., 2022) to alleviate communication cost. Consequently, a growing line of research has focused on collaborative fine-tuning under cross-domain data settings, referred to as *cross-domain federated fine-tuning (CD-FFT)*, which has demonstrated promising performance across diverse tasks (Yang et al., 2023; Feng et al., 2023; Su et al., 2024; Bai et al., 2024).

Backdoor attacks (Gu et al., 2017), a stealthy and potent threat to deep learning models, have already shown significant impact in FL (Xie et al., 2019; Bagdasaryan et al., 2020; Wang et al., 2020; Cheng et al., 2023; Zhang et al., 2023a; Nguyen et al., 2023; Liu et al., 2024). By compromising a subset of clients, adversaries can poison local updates and manipulate the global model to misclassify test samples containing specific triggers. Recent evidence further indicates that such attacks can effectively transfer to CD-FFT systems (Huang et al., 2024a). Due to the inherent domain discrepancies among clients and the rapid adaptability of fine-tuning, backdoor attacks can be injected into CD-FFT in a more covert and efficient manner, thereby raising substantial security concerns.

Existing backdoor defenses in FL can be broadly categorized into two classes: anomaly update detection (Cao et al., 2020; Rieger et al., 2022; Nguyen et al., 2022b; Zhang et al., 2022a; Xu et al., 2025) and robust aggregation (Blanchard et al., 2017; Yin et al., 2018; Fung et al., 2020; Pillutla et al., 2022; Huang et al., 2024b). These approaches have proven effective in traditional FL or class-imbalanced settings, primarily by measuring global gradient distances or employing coarse-grained control to suppress anomalous updates. However, gradient updates become more intricate due to domain shifts in CD-FFT, making such coarse-grained methods unreliable. They fail to capture the subtle backdoor behaviors that arise under domain discrepancies, and thus struggle to generalize to this more realistic scenario. This gap raises a key question: **Is it possible to propose an innovative defense mechanism for CD-FFT that enables fine-grained and robust backdoor mitigation?**

In response, we propose BEACON (**Be**havioral gr**a**dient de**c**oupling for cross-d**o**main federated fine-tu**n**ing), a novel backdoor defense framework tailored for CD-FFT. BEACON first creates *Task-Deviation Orthogonal Disentanglement* (TDOD) to project each client gradient onto the global task-consensus direction, thereby separating each update into task-aligned and deviation components. This orthogonal separation disentangles domain-specific variations from potentially malicious signals, enabling effective inter-client anomaly scoring. To further enhance defense capability, BEACON designs *Classification Head Inconsistency Forensics* (CHIF), which inspects abnormal behaviors in the fine-tuned classification head. By examining label-wise gradient patterns, CHIF identifies subtle manipulations in decision boundaries indicative of backdoor intentions. In conjunction, TDOD and CHIF empower BEACON to thwart backdoor attacks in an interpretable manner.

We evaluate BEACON on four cross-domain benchmarks, including DomainNet (Peng et al., 2019), PACS (Li et al., 2017), Office-Caltech-10 (Saenko et al., 2010) and OfficeHome (Venkateswara et al., 2017), as well as against three representative backdoor variants: BadNets (Gu et al., 2017), Neurotoxin (Zhang et al., 2022b), and contrastive backdoor injection (CBI) (Huang et al., 2024a). Extensive experiments validate the effectiveness of BEACON in CD-FFT, consistently suppressing attack success rates to below 2% while preserving main task accuracy with only negligible degradation. Our key contributions can be summarized as follows:

- We conduct the **first comprehensive evaluation** of transferring existing FL backdoor defenses to CD-FFT, and reveal their ineffectiveness stemming from reliance on coarse-grained criteria and neglect of domain discrepancies.

- To overcome these limitations, we propose BEACON, an innovative defense framework that decouples gradient behaviors at fine granularity, effectively bypassing domain discrepancies to expose backdoor injection and achieve robust backdoor thwarting.

- Unlike existing defenses that rely on coarse-grained similarity estimation, we design *Task-Deviation Orthogonal Disentanglement* to decompose each gradient into consensus and deviation components, enabling precise assessment of malicious behaviors. Furthermore, we propose *Classification Head Inconsistency Forensics* to capture intra-client boundary manipulations, thereby enhancing label-wise backdoor detection.

- Extensive experiments across four benchmarks and three backdoor variants demonstrate that BEACON achieves superior robustness and stability over SOTA FL defenses, consistently reducing attack success rates to below 2% while preserving main task performance.

## 2 RELATED WORK

### 2.1 CROSS-DOMAIN FEDERATED FINE-TUNING

Recent advances in FL have shifted from traditional settings that train models from scratch under class-imbalanced distributions (Wang et al., 2021; Park et al., 2023; Wei & Han, 2024; Le et al., 2024), toward more practical scenarios involving domain discrepancies across clients and parameter-efficient adaptation of pre-trained foundation models, referred to as CD-FFT systems. For instance, FedVPT (Yang et al., 2023) applies visual prompt tuning on each client and aggregates client-specific prompts at the server, offering a simple yet effective solution that has gained widespread adoption. FedIns (Feng et al., 2023) incorporates scaling-and-shifting feature (SSF) tuning (Lian et al., 2022) with SSF pools, enabling instance-level test-time adaptation through dynamic SSF vector selection. PromptFL (Li et al., 2023) distributes a frozen foundation model (e.g., CLIP

(Radford et al., 2021)) and collaboratively trains soft prompts across clients to achieve personalized adaptation. FedAPT (Su et al., 2024) further extends this line by using domain-specific keys to generate adaptive prompts during inference, built upon CLIP backbones.

## 2.2 BACKDOOR ATTACKS IN FEDERATED LEARNING

Bagdasaryan *et al.* (Bagdasaryan et al., 2020) proposed the first backdoor attack in FL by constraining and scaling malicious gradients to replace the global model with a backdoored version. Xie *et al.* (Xie et al., 2019) further developed a distributed backdoor attack, where multiple local triggers are combined to form a global trigger, thereby amplifying the overall backdoor effect and stealthiness. Neurotoxin (Zhang et al., 2022b) enhances stealth and persistence by injecting triggers into parameters that exhibit minimal updates across training rounds. Contrastive Backdoor Injection (CBI) (Huang et al., 2024a) extends these threats to CD-FFT by exploiting contrastive learning principles, marking the first dedicated study of backdoor vulnerabilities in this setting. By leveraging relationships between benign and poisoned samples, CBI significantly stregthens attack effectiveness. Therefore, these findings highlight the pronounced vulnerability of CD-FFT systems to backdoor attacks, underscoring the need for greater defensive attention.

## 2.3 BACKDOOR DEFENSES IN FEDERATED LEARNING

Existing defenses against backdoor attacks in FL can be broadly categorized into two groups: 1) anomaly update detection (Rieger et al., 2022; Nguyen et al., 2022b; Huang et al., 2024b; Xu et al., 2025), and 2) robust federated aggregation (Blanchard et al., 2017; Yin et al., 2018; Fung et al., 2020; Xie et al., 2021; Pillutla et al., 2022; Zhang et al., 2024).

**Anomaly Update Detection.** DeepSight (Rieger et al., 2022) identified neurons strongly correlated with backdoor behavior and applied HDBSCAN (McInnes et al., 2017) clustering to detect outliers. FLAME (Nguyen et al., 2022b) measured cosine similarity between client updates and introduced targeted noise to disrupt malicious gradients. AlignIns (Xu et al., 2025) proposed a direction alignment inspection mechanism that evaluates each client update based on its consistency with the global update direction and the sign alignment of parameters, thereby filtering out malicious updates that deviate from benign consensus.

**Robust Federated Aggregation.** Multi-Krum (Blanchard et al., 2017) selected client updates with minimal pairwise divergence to mitigate adversarial influence. FoolsGold (Fung et al., 2020) maintained historical update trajectories for each client and detected collusion through long-term similarity analysis. FLARE (Wang et al., 2022) leveraged latent representation distances to estimate client reliability and rank contributions accordingly.

Although these defenses have proven effective in conventional FL settings, they largely neglect the challenges introduced by domain shift and parameter-efficient fine-tuning. To fill this gap, our work designs fine-grained gradient decoupling methods tailored for CD-FFT, addressing this critical yet underexplored vulnerability.

## 3 PROBLEM FORMULATIONS

### 3.1 FEDERATED VISUAL PROMPT TUNING

We adopt federated visual prompt tuning (FedVPT) (Yang et al., 2023) as a representative framework to study CD-FFT system. We consider a FL system comprising a central server $\mathcal{S}$ and $N$ clients $\{C_1, C_2, \ldots, C_N\}$, where each client $C_i$ holds a private dataset $\mathcal{D}_i$ sampled from a specific domain. Each client fine-tunes a shared, frozen foundation model $\mathbf{F}$ by optimizing a small set of client-specific parameters $\boldsymbol{\delta}_i = \{\boldsymbol{\delta}_i^p, \boldsymbol{\delta}_i^h\}$, where $\boldsymbol{\delta}_i^p$ and $\boldsymbol{\delta}_i^h$ denote prompt and classifier head parameters, respectively.

During each communication round $t$, each client $C_i$ initializes its local tuning parameters from the global model $\boldsymbol{\delta}_G^t$, and obtains updated parameters $\boldsymbol{\delta}_i^t$ by minimizing the empirical loss on its private dataset:

$$\boldsymbol{\delta}_i^t = \arg\min_{\boldsymbol{\delta}_i} \mathbb{E}_{(x_i, y_i) \sim \mathcal{D}_i} \left[ \mathcal{L} \left( f(x_i; \mathbf{F}, \boldsymbol{\delta}_i), y_i \right) \right], \tag{1}$$

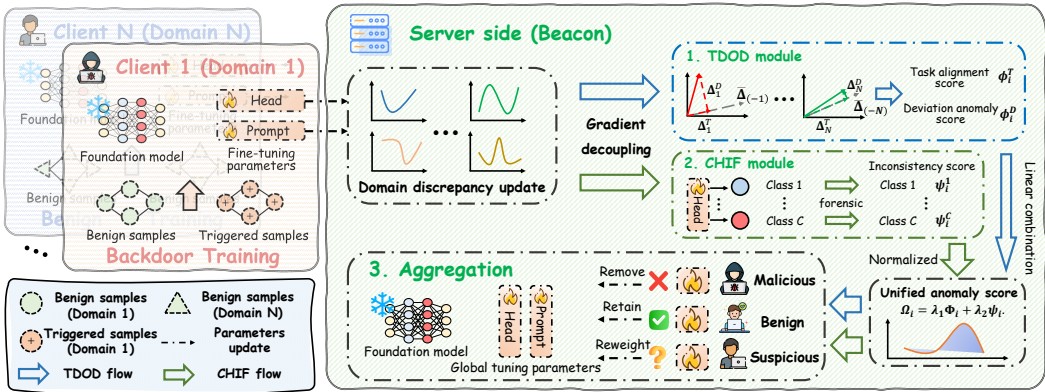

Figure 1: Illustration of the BEACON pipeline. We utilizes the **TDOD** and **CHIF** modules to compute fine-grained anomaly scores, which are subsequently integrated into a trust-aware aggregation strategy to defend against backdoor attacks and ensure the stability of the CD-FFT system.

where $\mathcal{L}$ denotes the classification loss and $f$ represents the inference function based on the backbone **F** and the learnable tuning parameters $\boldsymbol{\delta}$. The server then updates the global tuning parameters by aggregating the client updates as:

$$\boldsymbol{\delta}_G^{t+1} = \boldsymbol{\delta}_G^t + \frac{1}{N} \sum_{i=1}^{N} \boldsymbol{\Delta}_i^t. \tag{2}$$

Where $\boldsymbol{\Delta}_i^t = \boldsymbol{\delta}_i^t - \boldsymbol{\delta}_G^t$ denotes the local update, thus following the standard FedAVG (McMahan et al., 2017) scheme to synchronize client-side adaptations into the global model.

### 3.2 ATTACK AND DEFENSE MODEL

We assume an adversary $\mathcal{A}$ capable of compromising up to 50% of the clients, either concentrated within a single domain or distributed across multiple domains. These malicious clients can conduct both data poisoning and model poisoning to maximize the effectiveness and stealthiness of the attack. However, $\mathcal{A}$ has no knowledge of the benign clients' data distributions, cannot interfere with their training processes, and remains unaware of the defense strategies deployed on the server side.

The defender, i.e., the server $\mathcal{S}$, aims to detect and thwart backdoor attacks while preserving the utility and robustness of the CD-FFT system. Concretely, a practical defense should meet the following requirements: 1) substantially suppress the attack success rate to ensure system security; 2) maintain high accuracy on the main task to guarantee stability and cross-domain generalization; 3) operate without accessing clients' raw data or domain-specific distributions; and 4) avoid reliance on clean reference data or prior knowledge of poisoning triggers.

## 4 METHODOLOGY

### 4.1 OVERVIEW AND ADVANCES

The overall workflow of BEACON is illustrated in Alg. 1 and Fig. 1. Existing backdoor defenses in FL, when transferred to CD-FFT settings, suffer from severe limitations. They primarily rely on coarse-grained statistical anomaly detection in gradients, which becomes unreliable under domain-induced feature discrepancies and parameter-efficient fine-tuning frameworks. In such conditions, malicious manipulations are more subtle and entangled, preventing traditional defenses from generalizing to CD-FFT. To overcome these challenges, BEACON first decomposes each client update into a task-consensus component and a domain-deviation component, enabling precise assessment of suspicious behaviors. Moreover, it captures label-wise inconsistencies in classifier head activations, since malicious clients tend to manipulate specific decision boundaries to implant targeted backdoors. Finally, BEACON integrates these signals into trust scores for clients and performs robust trust-aware aggregation.

## 4.2 TASK-DEVIATION ORTHOGONAL DISENTANGLEMENT

We design the *Task-Deviation Orthogonal Disentanglement* (TDOD) module to decouple gradient behaviors. As a first step, we filter out inactive dimensions to reduce noise from trivial updates. Let each client update be denoted by $\mathbf{\Delta}_i \in \mathbb{R}^d$, we define the active dimension set as: $\Gamma = \{j \in [1, d] \mid \max_i |\mathbf{\Delta}_{i,j}| > \epsilon\}$, where $\epsilon$ is a small threshold.

To extract per-client consensus, we adopt a leave-one-out strategy to compute a task-consensus vector for each client based on the remaining population. Specifically, for client $i$, the task-consensus update is defined as:

$$\bar{\mathbf{\Delta}}_{(-i)} = \frac{1}{N-1} \sum_{k \neq i} \mathbf{\Delta}_k \in \mathbb{R}^{d'}, \tag{3}$$

where $d' = |\Gamma|$ denotes the dimensionality of the selected active index set. Each client's update $\mathbf{\Delta}_i$ is then decomposed into orthogonal components: the *task-alignment component* $\mathbf{\Delta}_i^T$ aligned with $\bar{\mathbf{\Delta}}_{(-i)}$, and the *domain-deviation component* $\mathbf{\Delta}_i^D$ orthogonal to it:

$$\mathbf{\Delta}_i^T = \frac{\langle \mathbf{\Delta}_i, \bar{\mathbf{\Delta}}_{(-i)} \rangle}{\|\bar{\mathbf{\Delta}}_{(-i)}\|^2} \cdot \bar{\mathbf{\Delta}}_{(-i)}, \tag{4}$$

$$\mathbf{\Delta}_i^D = \mathbf{\Delta}_i - \mathbf{\Delta}_i^T. \tag{5}$$

Here, $\langle \cdot, \cdot \rangle$ denotes the inner product between two vectors, and $\| \cdot \|$ represents the $\ell_2$ norm. Then, we define the task alignment score $\phi_i^T$ and deviation anomaly score $\phi_i^D$ as follows:

$$\phi_i^T = \frac{\|\mathbf{\Delta}_i^T\|}{\|\mathbf{\Delta}_i\|}, \quad \phi_i^D = \left| \frac{\|\mathbf{\Delta}_i^D\| - \mu_{S^D}}{\sigma_{S^D}} \right|, \tag{6}$$

where the and $\mu_{S^D}$ and $\sigma_{S^D}$ denote the mean and standard deviation of values given by $S^D = \{\|\mathbf{\Delta}_k^D\| \mid k = 1, \ldots, N\}$. Finally, we compute the overall task-deviation anomaly score as:

$$\Phi_i = (1 - \phi_i^T) + \phi_i^D, \tag{7}$$

which jointly captures task misalignment and excessive domain deviation for client $i$.

## 4.3 CLASSIFICATION HEAD INCONSISTENCY FORENSICS

We observe that backdoor behaviors often cause targeted perturbations in the classifier head, especially along specific label dimensions. These manipulations are designed to alter decision boundaries, resulting in label-wise deviations that differ significantly from benign client patterns.

Let the update vector of fine-tuned classifier head for client $i$ be denoted as $\mathbf{\Delta}_i^h \in \mathbb{R}^{C \times d_h}$, where $C$ is the number of classes and $d_h$ is the feature dimension of the classifier head. For each class $c \in \{1, \ldots, C\}$, we define the class-wise average update from all other clients as:

$$\mathbf{W}_{(-i)}^c = \frac{1}{N-1} \sum_{k \neq i} \mathbf{W}_k^c, \tag{8}$$

where $\mathbf{W}_k^c \in \mathbb{R}^{d_h}$ denotes the classifier head update vector for class $c$ in client $k$. To quantify potential tampering, we compute the classifier-head inconsistency score for each client $i$, which captures label-wise behavioral deviations from the population mean. Specifically, for each target class $c \in \{1, \ldots, C\}$, we define a forensic score:

$$\psi_i^c = -s(\mathbf{W}_i^c, \mathbf{W}_{(-i)}^c) + \frac{1}{C-1} \sum_{j \neq c} s(\mathbf{W}_i^j, \mathbf{W}_{(-i)}^j), \tag{9}$$

where $s(\mathbf{u}, \mathbf{v}) = \frac{\langle \mathbf{u}, \mathbf{v} \rangle}{\|\mathbf{u}\| \cdot \|\mathbf{v}\|}$ denotes the cosine similarity between two vectors. We compute the worst-case label-wise inconsistency as: $\tilde{\psi}_i = \max_{c \in \{1, \ldots, C\}} \psi_i^c$. Then, to ensure comparability across clients, We normalize the classifier-head inconsistency score as: $\psi_i = \tilde{\psi}_i / \left\| \left\{ \tilde{\psi}_j \right\}_{j=1}^N \right\|$.

A high value of $\psi_i$ indicates suspicious concentration of classifier head updates toward a specific class, which is characteristic of targeted backdoor injection.

## 4.4 ANOMALY DETECTION AND AGGREGATION DECISION

We define a unified anomaly score $\Omega_i$ for each client $C_i$ by linearly combining the task-deviation score $\Phi_i$ and the inconsistency forensic score $\psi_i$:

$$\Omega_i = \lambda_1 \Phi_i + \lambda_2 \psi_i, \quad \lambda_1 + \lambda_2 = 1. \quad (10)$$

Based on the anomaly score $\Omega_i$, each client is assigned a trust category: *benign*, *suspicious*, or *malicious*, and corresponding aggregation weights are applied. This classification ensures that malicious clients are excluded, while the influence of suspicious clients is attenuated. The final aggregation weight $w_i$, which replaces the uniform weight $1/N$ for client $i$, is defined as:

$$w_i = \begin{cases} 0, & \Omega_i > \tau_m \\ 1 - \dfrac{\Omega_i}{\|\Omega_{\mathcal{B} \cup \mathcal{S}}\|}, & \tau_s < \Omega_i \leq \tau_m \quad (11) \\ 1, & \Omega_i \leq \tau_s, \end{cases}$$

where $\tau_s$ and $\tau_m$ denote the thresholds for suspicious and malicious clients, respectively. The set $\Omega_{\mathcal{B} \cup \mathcal{S}}$ contains the anomaly scores of all non-malicious clients, and $\|\cdot\|$ represents the $\ell_2$ norm.

This aggregation strategy enables robust model updates by strictly filtering high-risk clients and proportionally reducing the influence of moderately abnormal ones, while maintaining the contributions of benign participants.

---

**Algorithm 1: BEACON**

**Input:** Client updates $\{\mathbf{\Delta}_i\}_{i=1}^N$, classifier heads $\{\mathbf{\Delta}_i^h\}_{i=1}^N$, thresholds $\tau_s, \tau_m$, weights $\lambda_1, \lambda_2$

**Output:** Aggregated update $\mathbf{\Delta}_G$

**foreach** *client* $i \in \{1, \dots, N\}$ **do**
  Compute $\Phi_i \leftarrow \text{TDOD}(\mathbf{\Delta}_i, \bar{\mathbf{\Delta}}_{(-i)})$;
  Compute $\psi_i \leftarrow \text{CHIF}(\mathbf{\Delta}_i^h)$;
  Compute anomaly score:
  $\Omega_i \leftarrow \lambda_1 \Phi_i + \lambda_2 \psi_i$;
**end**
Initialize $\mathcal{B} \leftarrow \emptyset, \mathcal{S} \leftarrow \emptyset$;
**foreach** *client* $i$ **do**
  **if** $\Omega_i > \tau_m$ **then**
    $w_i \leftarrow 0$ ;        // Malicious
  **else if** $\Omega_i > \tau_s$ **then**
    $\mathcal{S} \leftarrow \mathcal{S} \cup \{i\}$ ;  // Suspicious
  **else**
    $\mathcal{B} \leftarrow \mathcal{B} \cup \{i\}$;      // Benign
  **end**
**end**
**foreach** *client* $i$ **do**
  **if** $i \in \mathcal{B}$ **then**
    $w_i \leftarrow 1$
  **else if** $i \in \mathcal{S}$ **then**
    $w_i \leftarrow 1 - \Omega_i/\|\{\Omega_j\}_{j \in \mathcal{B} \cup \mathcal{S}}\|$
**end**

Aggregate global update: $\mathbf{\Delta}_G \leftarrow \dfrac{\sum_{i=1}^N w_i \cdot \mathbf{\Delta}_i}{N}$
**return** $\mathbf{\Delta}_G$

---

## 5 EXPERIMENTS

In this section, we comprehensively evaluate the transferability of existing FL defenses to CD-FFT and demonstrate the superior performance of our proposed BEACON framework in suppressing backdoors, preserving main task accuracy, and maintaining system stability. We further provide fine-grained domain-wise results and ablation studies to assess the contributions of different modules. In addition, we extend the analysis with runtime overhead evaluation (Appendix F), robustness under dynamic attacks (Appendix G), and visualization of anomaly scores (Appendix I), offering a holistic validation of BEACON.

### 5.1 EXPERIMENTAL SETUP

BEACON is deployed on a server with an Intel(R) Xeon(R) Platinum 8259CL CPU @ 2.50GHz CPU, 128GB RAM. Our experiments are implemented using Pytorch and conducted on an NVIDIA RTX 3090 GPU.

**Datasets and Model Settings.** We conduct experiments on four widely used cross-domain benchmarks: **PACS** (Li et al., 2017), **DomainNet** (Peng et al., 2019), **Office-Caltech10** (Saenko et al., 2010), and **Office-Home** (Venkateswara et al., 2017). We adopt **ViT-Base/16** (Dosovitskiy et al., 2020) as the backbone model and apply **visual prompt tuning** (Jia et al., 2022) for client adaptation. Detailed dataset descriptions and model configurations are provided in the Appendix C.

**Evaluation Metrics.** Following AlignIns (Xu et al., 2025), we evaluate defense performance using three metrics. **Main Task Accuracy (MTA)** is the standard classification accuracy on clean test samples across all domains. **Attack Success Rate (ASR)** is the fraction of samples with triggers that are classified into the target class specified by the adversary, and **ASR reduction** denotes the change in ASR relative to the attack-only (no defense) setting. **Robust Accuracy (RA)** is the fraction of test

| Methods | DomainNet | | | PACS | | | Office-Home | | | Office-Caltech-10 | | |
|---|---|---|---|---|---|---|---|---|---|---|---|---|
| | $MTA$ | $ASR$ | $RA$ | $MTA\uparrow$ | $ASR\downarrow$ | $RA\uparrow$ | $MTA\uparrow$ | $ASR\downarrow$ | $RA\uparrow$ | $MTA\uparrow$ | $ASR\downarrow$ | $RA\uparrow$ |
| *Attack with Badnets* | | | | | | | | | | | | |
| Clean | 84.21 | 2.63 | 84.19 | 95.30 | 1.90 | 95.11 | 85.97 | 1.86 | 86.13 | 94.86 | 1.18 | 94.86 |
| No defense | 84.59 | 92.23 | 7.33 | 94.64 | 85.41 | 14.37 | 89.05 | 90.89 | 8.48 | 95.60 | 83.86 | 15.88 |
| T-Mean | **84.42** | 93.65 | 5.92 | 94.35 | 80.20 | 19.51 | 87.34 | 88.85 | 10.54 | 95.53 | 70.46 | 29.14 |
| Median | 84.28 | 93.38 | 6.20 | **94.35** | 80.89 | 18.81 | 88.07 | 87.33 | 11.60 | 94.15 | 70.35 | 29.25 |
| Foolsgold | 76.51 | 3.17 | 76.22 | 82.21 | 18.12 | 79.29 | 90.06 | 1.40 | 89.92 | 97.71 | 0.13 | 97.71 |
| RFA | 37.18 | 47.39 | 36.00 | 66.23 | 28.25 | 65.12 | 80.44 | 16.27 | 78.82 | 92.66 | 0.13 | 92.66 |
| FLAME | 82.11 | 30.54 | 63.76 | 90.67 | **4.97** | 90.47 | 91.86 | 2.10 | 91.89 | 97.87 | 0.13 | 97.13 |
| Deepsight | 84.16 | 93.33 | 6.28 | 93.98 | 74.42 | 25.28 | 88.95 | 87.11 | 12.27 | 96.26 | 28.73 | 70.08 |
| AlignIns | 84.03 | 92.30 | 7.38 | 88.55 | 29.67 | 68.74 | 92.86 | 0.52 | 92.68 | **98.34** | 0.14 | **98.34** |
| *Beacon* | 83.89 | **1.34**$_{\downarrow1.83}$ | **93.68**$_{\uparrow17.46}$ | 92.81 | 6.00 | **91.34**$_{\uparrow0.87}$ | **92.93**$_{\uparrow0.07}$ | 0.52 | **92.83**$_{\uparrow0.15}$ | 98.23 | **0.00**$_{\downarrow0.13}$ | 98.23 |
| *Attack with Neurotoxin* | | | | | | | | | | | | |
| Clean | 84.21 | 2.63 | 84.19 | 95.30 | 1.90 | 95.11 | 85.97 | 1.86 | 86.13 | 94.86 | 1.18 | 94.86 |
| No defense | 84.21 | 89.82 | 9.64 | 94.71 | 84.80 | 15.05 | 87.87 | 82.70 | 16.24 | 95.73 | 85.65 | 14.22 |
| T-Mean | 84.38 | 93.00 | 6.50 | 94.95 | 83.76 | 16.17 | 89.53 | 90.43 | 9.22 | 95.57 | 63.12 | 35.06 |
| Median | **84.74** | 93.51 | 6.13 | **94.99** | 84.15 | 15.77 | 89.70 | 90.16 | 9.49 | 95.58 | 61.37 | 36.81 |
| Foolsgold | 78.24 | 6.43 | 78.41 | 86.99 | 7.97 | 84.98 | 74.30 | **0.70** | 74.47 | 97.44 | 0.00 | 97.30 |
| RFA | 71.45 | 16.04 | 70.99 | 73.38 | 15.80 | 72.62 | 82.95 | 4.91 | 82.16 | 91.48 | 0.26 | 91.95 |
| FLAME | 80.95 | 4.22 | 80.91 | 85.68 | 6.17 | 84.86 | 90.41 | 1.43 | 89.96 | 98.50 | 0.00 | 98.50 |
| Deepsight | 83.98 | 92.86 | 6.69 | 94.50 | 80.43 | 19.35 | 88.47 | 90.82 | 7.94 | 96.78 | 88.63 | 14.36 |
| AlignIns | 83.64 | 97.77 | 1.68 | 93.35 | 74.51 | 25.11 | 93.78 | 0.87 | 92.98 | 98.11 | 0.00 | 98.11 |
| *Beacon* | 83.60 | **1.27**$_{\downarrow2.95}$ | 83.86$_{\downarrow2.95}$ | 93.66 | **5.14**$_{\downarrow1.03}$ | 86.35$_{\uparrow1.37}$ | **94.15**$_{\uparrow0.37}$ | 1.26 | **94.08**$_{\uparrow1.10}$ | **98.77**$_{\uparrow0.22}$ | 0.00 | **98.64**$_{\uparrow0.14}$ |
| *Attack with CBI* | | | | | | | | | | | | |
| Clean | 84.21 | 2.63 | 84.19 | 95.30 | 1.90 | 95.11 | 85.97 | 1.86 | 86.13 | 94.86 | 1.18 | 94.86 |
| No defense | 84.14 | 96.18 | 3.29 | 94.07 | 87.29 | 12.41 | 88.69 | 93.92 | 5.63 | 96.40 | 95.13 | 4.74 |
| T-Mean | 83.70 | 97.72 | 2.03 | 94.33 | 94.93 | 4.99 | 89.66 | 94.18 | 5.65 | 95.61 | 93.65 | 6.08 |
| Median | 83.15 | 95.68 | 3.98 | 94.64 | 94.32 | 5.60 | 90.17 | 94.83 | 5.00 | 94.25 | 94.12 | 5.60 |
| Foolsgold | 80.16 | 4.59 | 79.10 | 83.76 | 17.68 | 75.99 | 88.28 | **0.52** | 88.39 | 96.88 | 0.00 | 96.90 |
| RFA | 57.74 | 25.97 | 59.17 | 80.02 | 13.11 | 77.98 | 62.94 | 29.76 | 61.73 | 40.26 | 51.55 | 39.87 |
| FLAME | 9.73 | 88.18 | 9.41 | 90.68 | 5.01 | 80.61 | 91.83 | 1.05 | **91.31** | 97.59 | 0.00 | 97.72 |
| Deepsight | 84.04 | 97.57 | 2.14 | 94.29 | 91.24 | 8.53 | 87.89 | 95.34 | 4.15 | 96.53 | 94.56 | 5.30 |
| AlignIns | 83.23 | 95.56 | 4.06 | 93.43 | 2.14 | 93.45 | **94.88** | 97.43 | 2.23 | 98.35 | 0.00 | 98.35 |
| *Beacon* | **84.73**$_{\uparrow0.69}$ | **1.65**$_{\downarrow2.94}$ | **84.92**$_{\uparrow5.82}$ | **94.91**$_{\uparrow0.27}$ | 2.14 | **94.98**$_{\uparrow0.53}$ | 93.91 | 3.45 | 89.44 | **98.50**$_{\uparrow0.15}$ | 0.00 | **98.50**$_{\uparrow0.15}$ |

Table 1: **Comparison between BEACON and baselines across four cross-domain datasets under three backdoor variants.** Bold values indicate the best performance among all defense methods. ↑ and ↓ markers denote the relative improvement or degradation of BEACON compared to the second-best defense method. All results are reported in percentage format (%).

samples with triggers that are still correctly classified into their true source class. An ideal defense should maintain a high **MTA** and **RA**, while simultaneously reducing **ASR**.

**Attack Settings and Defense Baselines.** We consider three attack strategies: **BadNet** (Gu et al., 2017), **Neurotoxin** (Zhang et al., 2022b), and **Contrastive Backdoor Injection (CBI)** (Huang et al., 2024a). We compare BEACON with **Trimmed Mean (T-Mean)** (Yin et al., 2018), **Median** (Yin et al., 2018), **FoolsGold** (Fung et al., 2020), **FLAME** (Nguyen et al., 2022b), **RFA** (Pillutla et al., 2022), **DeepSight** (Rieger et al., 2022), and **AlignIns** (Xu et al., 2025). We adopt the default settings for defense baselines, and integrate them into the CD-FFT system. Detailed descriptions of the attack implementations and comparison defenses are provided in Appendix D and Appendix E.

**Implementation Details.** Before initiating any backdoor attack, we pretrain the global model on each dataset for 50 rounds. We adopt a default setting of 3 clients per domain. In the main experiments, we designate client 3 and client 6 as the malicious participants. During backdoor training, the poisoning local epoch is set to 2, the poisoning learning rate to 0.01, and the default poisoned data ratio $\gamma = 0.25$. The total number of communication rounds is 80. For our BEACON, we use $\lambda_1 = 0.6$ and $\lambda_2 = 0.4$ by default. The thresholds for suspicious and malicious classification are set to $\tau_s = 0.4$ and $\tau_m = 0.8$, respectively. The local training batch size for all clients is fixed at 32.

## 5.2 MAIN RESULTS

**Backdoor Suppression.** We first evaluate the effectiveness of our framework and baselines across four cross-domain datasets under three backdoor variants. As shown in Tab. 1, BEACON consistently suppresses the ASR to below **2%** across nearly all settings. For instance, on the Office-Caltech10 dataset, BEACON reduces the ASR of all three attacks to **0%**, indicating a complete thwarting of the backdoor effect. In contrast, existing defenses such as **Trimmed Mean** (Yin et al., 2018), **Median** (Yin et al., 2018), and **DeepSight** (Rieger et al., 2022) fail almost entirely across all datasets. Their ineffectiveness under the CD-FFT setting stems from their inability to adapt to domain shifts

and the partial fine-tuning paradigm. While **FoolsGold** (Fung et al., 2020) demonstrates backdoor suppression in some cases, it exhibits poor stability on the main task and fails to eliminate backdoor behavior on PACS. Moreover, **AlignIns** (Xu et al., 2025) and **FLAME** (Nguyen et al., 2022b) perform inadequately on DomainNet, likely due to the complex feature distribution and severe domain shifts. These characteristics render coarse-grained defenses insufficiently expressive to capture subtle backdoor signals.

**Main Task and Robust Accuracy.** As shown in Tab. 1, BEACON consistently achieves the highest or near-highest RA across all datasets and attacks. This indicates that BEACON preserves strong accuracy even on poisoned inputs, closely matching the behavior of the clean model. Regarding MTA, BEACON demonstrates remarkable stability across all settings. While BEACON may not attain the highest MTA in every case, such as its 83.89 on DomainNet under the BadNets attack compared to 84.42 by **Trimmed Mean**, we find that methods with marginally higher MTA often fail to effectively defend against backdoors. Although **AlignIns** successfully mitigates ASR in some scenarios, its MTA and RA remain consistently lower than those of our BEACON.

Overall, BEACON maintains both strong main task performance and robust generalization, outperforming all baselines and supporting stable operation in the CD-FFT setting.

**Fine-grained Domain Results.** We present domain-wise results on DomainNet, PACS, and Office-Home in Fig. 2. Here, $\mathcal{C}$ denotes the *Clipart* domain, with the remaining symbols corresponding to other domains in the datasets. As shown, BEACON consistently achieves superior performance across domains, especially on DomainNet and PACS. For instance, in the *Painting* domain of DomainNet, BEACON reduces ASR by **92.35%** while maintaining

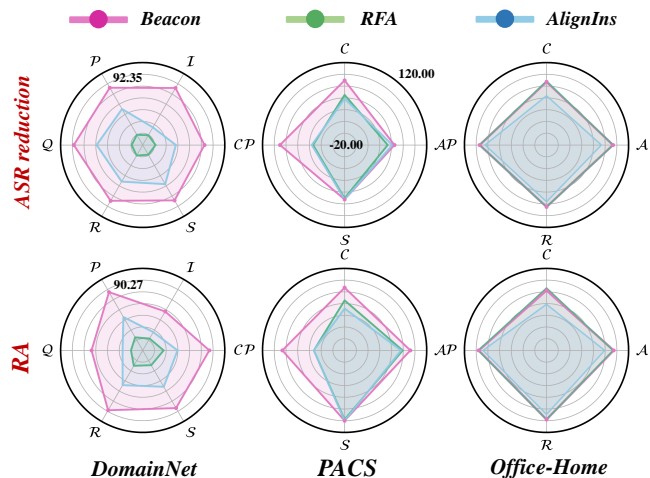

Figure 2: Comparisons of BEACON and baselines in thwarting ASR and maintaining RA across each fine-grained domain.

a high RA of **90.27%**. In contrast, methods such as RFA and AlignIns struggle to balance ASR suppression and RA preservation, and in some domains, they fail to mitigate the backdoor effects at all. BEACON achieves consistent and robust defense performance across diverse domains.

### 5.3 ABLATION STUDIES

**Necessity of TDOD and CHIF Modules.** Tab. 2 presents the ablation results of BEACON with only the TDOD or the CHIF module under **BadNets** attack on DomainNet and PACS. We observe that when equipped with only the CHIF module, BEACON occasionally fails to suppress the backdoor. For instance, on DomainNet, the ASR is only reduced by approximately half (from 92.23% to 48.82%). In contrast, when using only the TDOD module, the main task performance degrades significantly, as evidenced by an MTA of only 89.01% on PACS. By integrating both modules, BEACON consistently achieves strong performance, effectively eliminating backdoors while maintaining high MTA and RA. These results demonstrate the necessity of combining TDOD and CHIF for robust and stable backdoor defense.

**Number of Total Clients.** We investigate the robustness of BEACON under varying numbers of clients per domain, ranging from 1 to 5. As shown in Fig. 3, BEACON maintains strong defense capabilities even under extreme conditions. Notably, when there is only one client per domain, which means that nearly 50% of the participating clients are malicious, BEACON still reduces the ASR by **96.29%** on the PACS dataset. It is worth noting that when each domain contains two clients, BEACON does not achieve the absolute best ASR reduction, suppressing approximately 70% of the attack success. Nevertheless, this still represents a significant mitigation of backdoor effects,

| TDOD | CHIF | DomainNet | | | PACS | | |
|---|---|---|---|---|---|---|---|
| | | MTA | ASR | RA | MTA | ASR | RA |
| ✗ | ✗ | 84.59 | 92.23 | 7.33 | 94.64 | 85.41 | 14.37 |
| ✗ | ✓ | 83.46 | 48.82 | 49.12 | **93.87** | 13.27 | 84.06 |
| ✓ | ✗ | 83.33 | 25.10 | 66.50 | 89.01 | 15.18 | 77.28 |
| ✓ | ✓ | **83.89** | **1.34** | **93.68** | 92.81 | **6.00** | **91.34** |

Table 2: Ablation Results on TDOD and CHIF Components. BEACON equipped with both TDOD and CHIF consistently achieves the best ASR suppression and RA performance, while maintaining high stability.

| Dataset&Metrics | Poisoning Ratio | | | | | | |
|---|---|---|---|---|---|---|---|
| | 0.125 | 0.250 | 0.375 | 0.500 | 0.625 | 0.750 | 0.875 |
| *DomainNet*, Attack domain: $\mathcal{I}, \mathcal{P}$ | | | | | | | |
| MTA↑ | 83.73 | 83.89 | 83.02 | 84.64 | **84.65** | 82.48 | 82.72 |
| ASR↓ | 2.14 | 1.34 | **1.29** | 3.04 | 21.60 | 36.62 | 2.59 |
| RA↑ | 83.96 | 84.07 | 83.32 | **84.74** | 71.71 | 59.92 | 83.01 |
| *Office-Home*, Attack domain: $\mathcal{C}, \mathcal{P}$ | | | | | | | |
| MTA↑ | 92.46 | 92.87 | 93.92 | **94.67** | 91.82 | 90.64 | 89.49 |
| ASR↓ | **0.52** | 0.87 | 1.05 | 3.66 | 1.92 | 4.53 | 4.26 |
| RA↑ | 92.46 | 92.70 | **93.12** | 92.11 | 91.86 | 90.19 | 90.04 |

Table 3: Ablation results on poisoning ratios. BEACON accurately identifies backdoor attacks across all poisoning ratios.

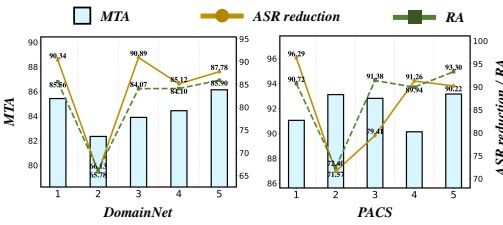

Figure 3: BEACON's stability under varying numbers of participating clients per domain. BEACON accurately identifies backdoor behaviors and maintains system stability in both **large-scale** and **small-scale** client scenarios.

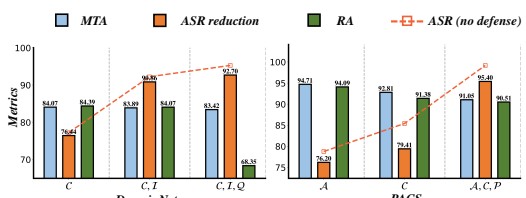

Figure 4: BEACON's performance under varying numbers of malicious domains. Even **multi-domain collaborative attacks** cannot evade Beacon's precise defense.

indicating that BEACON remains highly effective in alleviating attack pressure across various client participation scenarios.

**Number of Malicious Domains.** To evaluate the robustness of BEACON against **collaborative backdoor attacks** across multiple domains, we vary the number of malicious domains controlled by the attacker. As shown in Fig. 4, BEACON consistently mitigates the backdoor effect on both DomainNet and PACS datasets, reducing the ASR to nearly zero in most cases. When all three domains in DomainNet contain malicious clients, the system's RA slightly drops to 68.35. Nevertheless, the MTA remains relatively high, indicating that the global model still retains task utility despite the challenging setting.

**Poisoning Ratio Sensitivity.** We investigate the sensitivity of BEACON to different poisoning ratios by varying the proportion of poisoned data in each local batch from 0.125 to 0.875. Tab. 3 reports the results on the DomainNet and Office-Home datasets, demonstrating that BEACON exhibits strong resilience across a wide range attack settings. A notable exception occurs on DomainNet when the poisoning ratio is set to 0.625 or 0.750, where the ASR is not suppressed to the lowest possible level. Nevertheless, an ASR of only 21.6% still reflects substantial mitigation.

Overall, BEACON demonstrates consistently robust performance under diverse and challenging configurations, validating its generalization and reliability.

## 6 CONCLUSION

In this paper, we conduct the first comprehensive evaluation of transferring existing FL backdoor defenses to CD-FFT settings and propose BEACON, the first defense framework tailored for this challenging scenario. BEACON decouples gradient behaviors through the *Task-Deviation Orthogonal Disentanglement (TDOD)* and *Classification Head Inconsistency Forensics (CHIF)* modules, enabling fine-grained identification of malicious updates. We further integrate the anomaly signals into a trust-aware aggregation to maintain system stability while suppressing backdoor threats. Extensive experiments across four cross-domain datasets and three backdoor variants show that **BEACON reduces ASR to below 2% in almost all cases**, while preserving high MTA and RA. Compared to seven state-of-the-art defenses, BEACON achieves the most consistent and stable performance under various attack configurations.

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

## A    APPENDIX OUTLINE

This appendix is organized as follows:

- **Sec. B** clarifies that large language models usage.
- **Sec. C** provides additional dataset and model details, including domain statistics, class selections, and prompt tuning configurations.
- **Sec. D** describes the implementation details of the three representative backdoor attacks (BadNets, Neurotoxin, and CBI) and illustrates the trigger settings used in our experiments.
- **Sec. E** summarizes the defense baselines considered in our comparisons and clarifies their implementation settings in CD-FFT.
- **Sec. F** reports the runtime analysis of the TDOD and CHIF modules, showing their negligible overhead and scalability with respect to client numbers.
- **Sec. G** investigates BEACON's robustness under dynamic and colluding adversaries, including constrain-and-scale based model replacement attacks.
- **Sec. H** discusses the limitations of the current framework and outlines directions for future extensions.
- **Sec. I** presents additional visualization results of anomaly scores across different datasets and attack variants.

## B    LLM USAGE STATEMENT

During the preparation of this manuscript, the authors used OpenAI ChatGPT-5 for proofreading and improving the readability of the text. The LLM was not involved in generating technical content, research ideas, or experimental results. Following its use, the authors carefully reviewed and revised all outputs, and take full responsibility for the final publication.

## C    DATASET AND MODEL DETAILS

The details of the four cross-domain datasets used in our experiments are as follows.

- **DomainNet** (Peng et al., 2019) is a large-scale benchmark that comprises 345 object categories spanning six diverse domains, namely *Clipart*, *Infograph*, *Painting*, *Quickdraw*, *Real*, and *Sketch*. Following Li et al. (2020), we select ten representative classes to conduct our experiments.
- **PACS** (Li et al., 2017) consists of 7 object categories distributed across four visually distinct domains, including *Artpainting*, *Cartoon*, *Photo*, and *Sketch*, which are widely used for evaluating cross-domain generalization.
- **Office-Home** (Venkateswara et al., 2017) contains 65 object categories collected from four domains with significant appearance shifts, namely *Art*, *Clipart*, *Product*, and *Realworld*. In our study, we utilize the first ten categories for evaluation.
- **Office-Caltech10** (Saenko et al., 2010) is a domain adaptation benchmark that shares 10 common object categories across four domains, including *Amazon*, *DSLR*, *Webcam*, and *Caltech*.

We use **ViT-Base/16** (Dosovitskiy et al., 2020) as the backbone. Each client adopts **visual prompt tuning** (Jia et al., 2022) in the VPT-Deep configuration: the ViT backbone is frozen and we update only the inserted prompt tokens and the task-specific classification head. Concretely, we insert 15 prompt tokens (dimension 768) at each of the 12 transformer blocks.

## D    DETAILS OF THE BACKDOOR ATTACK IMPLEMENTATIONS

We summarize the three backdoor attacks used in our experiments.

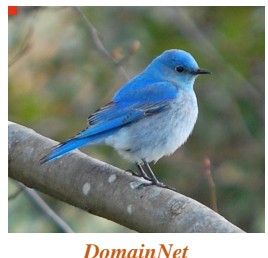 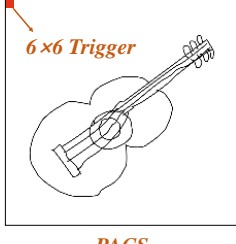 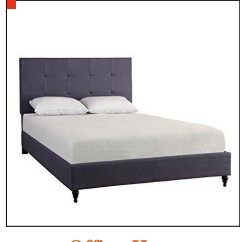 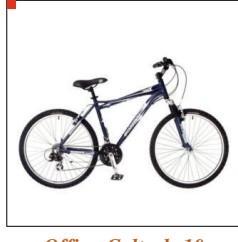

*DomainNet*
(*Real domain*)

*PACS*
(*Sketch domain*)

*Office-Home*
(*Product domain*)

*Office-Caltech-10*
(*Amazon domain*)

Figure 5: Examples from the four datasets and visualizations of trigger patterns.

**BadNets.** We use the standard square-patch BadNets (Gu et al., 2017) with a fixed location and color. This patch-based attack serves as our baseline.

**Neurotoxin.** Following (Zhang et al., 2022b), we inject parameter-level perturbations into the least-active parameters. Concretely, we target parameters in the bottom 98% percentile of per-parameter update magnitudes to maximize stealthiness.

**Contrastive Backdoor Injection (CBI).** We implement CBI (Huang et al., 2024a) as a model-poisoning method that combines cross-entropy with a contrastive loss. The local objective is $\mathcal{L} = \mathcal{L}_{\text{CE}} + 0.5 \cdot \mathcal{L}_{\text{contrast}}$ (contrastive weight = 0.5).

Fig. 5 illustrates the trigger patterns and example poisoned samples for the four datasets.

# E DETAILS OF THE COMPARISON DEFENSE METHODS

We briefly summarize the baseline methods included in our experiments.

**FedVPT (Clean)** (Yang et al., 2023). In the clean setting, we adopt federated visual prompt tuning, where each client fine-tunes only the inserted prompt tokens and the classification head, while the server aggregates updated parameters using FedAvg.

**Krum** (Blanchard et al., 2017). Krum selects a single client update that is closest to others in terms of Euclidean distance, thereby filtering out potential outliers. This makes it resilient to a small number of malicious updates but less scalable when many clients are compromised.

**Trimmed Mean and Median** (Yin et al., 2018). These aggregation rules reduce the influence of extreme updates. The trimmed mean discards a fixed proportion of the largest and smallest parameter values before averaging, while the coordinate-wise median takes the middle value for each parameter dimension.

**FoolsGold** (Fung et al., 2020). FoolsGold monitors historical update directions and downweights clients with overly similar updates, under the assumption that colluding adversaries submit aligned gradients to reinforce the backdoor. By reducing their contribution, it limits the amplification effect of coordinated malicious clients.

**RFA** (Pillutla et al., 2022). RFA applies the geometric median to aggregate client updates, which minimizes the overall $\ell_2$ distance to all updates. This robust estimator diminishes the effect of extreme values, making the global model less sensitive to poisoned contributions.

**FLAME** (Nguyen et al., 2022b). FLAME is a defense framework against backdoor attacks that injects carefully calibrated noise to disrupt malicious behavior while preserving accuracy on clean data. It combines model clustering with weight pruning to minimize the required noise, offering strong robustness with little sacrifice of utility.

**DeepSight** (Rieger et al., 2022). DeepSight identifies neurons strongly associated with backdoor behavior and leverages HDBSCAN clustering to detect and isolate outlier updates. This neuron-level perspective enables more precise anomaly detection than coarse-grained statistics.

**AlignIns** (Xu et al., 2025). AlignIns inspects the directional alignment of client updates by measuring their consistency with the global update direction and the sign alignment of parameters. Updates that deviate significantly from the benign consensus are flagged as suspicious and suppressed.

## F  RUNTIME ANALYSIS OF TDOD AND CHIF

We report the empirical time overhead of the two BEACON modules (TDOD and CHIF), and evaluate scalability by varying the number of clients per domain. Tab. 4 summarizes the measured times: for each dataset we list the average local training time observed for a malicious client and a benign client, and the cost of TDOD and CHIF across 1–5 clients per domain.

Two key observations follow. First, the absolute cost of TDOD is tiny: across all datasets TDOD incurs on the order of $10^{-3}$–$10^{-2}$ seconds (e.g., up to $\approx 0.0185$s on DomainNet for 5 clients per domain). CHIF is more expensive but still lightweight, ranging from a few $10^{-3}$s up to a few $10^{-1}$s depending on dataset and client count (DomainNet shows the largest CHIF cost, $\approx 0.3776$s at 5 clients per domain). By contrast, a single round of local training on a selected client takes on the order of 1–5 seconds in our measurements (see the table headers), so the combined TDOD+CHIF overhead is negligible relative to local training time.

Second, the per-round inspection cost grows approximately linearly with the number of inspected clients: doubling the number of clients roughly doubles the total inspection time, consistent with $O(N)$ complexity with respect to the number of participating clients. This linear behaviour, together with the very small constant for TDOD, supports the practical scalability of BEACON when deployed on the server side.

In short, BEACON adds only a few seconds of extra work per round in realistic settings. The small and linear overheads confirm that BEACON can be deployed without materially slowing down federated training while providing robust, fine-grained backdoor detection.

| Module | Number of client per domain | | | | |
|---|---|---|---|---|---|
| | 1 | 2 | 3 | 4 | 5 |
| *DomainNet, Malicious training: 4.3297s, Benign training: 2.0654s* | | | | | |
| TDOD | 0.0081 | 0.0111 | 0.0114 | 0.0167 | 0.0185 |
| CHIF | 0.0775 | 0.1550 | 0.2278 | 0.3088 | 0.3776 |
| *Office-Home, Malicious training: 3.5464s, Benign training: 1.3342s* | | | | | |
| TDOD | 0.0037 | 0.0068 | 0.0075 | 0.0114 | 0.0140 |
| CHIF | 0.0542 | 0.0972 | 0.1511 | 0.1947 | 0.2732 |

| Module | Number of client per domain | | | | |
|---|---|---|---|---|---|
| | 1 | 2 | 3 | 4 | 5 |
| *PACS, Malicious training: 3.3830s, Benign training: 1.6895s* | | | | | |
| TDOD | 0.0036 | 0.0051 | 0.0099 | 0.0116 | 0.0118 |
| CHIF | 0.0292 | 0.0506 | 0.0761 | 0.1015 | 0.1288 |
| *Office-Caltech-10, Malicious training: 1.0419s, Benign training: 0.8969s* | | | | | |
| TDOD | 0.0029 | 0.0049 | 0.0073 | 0.0093 | 0.0125 |
| CHIF | 0.0487 | 0.0963 | 0.1445 | 0.1941 | 0.2435 |

Table 4: Runtime analysis of TDOD and CHIF across four datasets with varying numbers of clients per domain. The overhead remains negligible compared to local training time, and scales linearly with the number of clients, demonstrating the efficiency and scalability of BEACON.

## G  ROBUSTNESS UNDER DYNAMIC ATTACKS

We test BEACON in a stronger, dynamic threat model where adversaries combine Neurotoxin (Zhang et al., 2022b) and a constrain-and-scale attack (Bagdasaryan et al., 2020). Our experimental setup uses two colluding malicious clients and three clients per domain. Tab. 5 reports the resulting MTA, ASR and RA across the four datasets.

Above all, BEACON consistently suppresses ASR and preserves RA in the majority of cases: on DomainNet, Office-Home and Office-Caltech-10 we observe substantial ASR reduction and near-clean RA, indicating that BEACON effectively exposes stealthy manipulations. Second, PACS is a partial failure mode: under dynamic attacks malicious updates are sometimes labeled as *suspicious* rather than *malicious* and therefore receive attenuated but non-zero weights. Over multiple rounds these small contributions can gradually restore the backdoor. This behavior stems from the subtle, multi-round accumulation strategy.

Importantly, this PACS failure mode is controllable. Tightening the thresholds $\tau_m$ and $\tau_s$ reduces false negatives (i.e., suspicious updates that should be blocked) at the cost of slightly more conservative filtering. Overall, the experiments demonstrate that BEACON is robust to dynamic, coordinated

attacks in practical settings, and that any remaining corner cases can be addressed by small, interpretable changes to the detection thresholds.

| Method | DomainNet | | | PACS | | |
|---|---|---|---|---|---|---|
| | $MTA$ | $ASR$ | $RA$ | $MTA$ | $ASR$ | $RA$ |
| Clean | 84.21 | **2.63** | 84.19 | **95.30** | **1.90** | **95.11** |
| No defense | 84.36 | 97.76 | 1.89 | 94.93 | 83.74 | 16.03 |
| *Beacon* | **84.43** | 2.65 | **84.25** | 94.85 | 44.43 | 55.13 |
| Method | Office-Home | | | Office-Caltech-10 | | |
| | $MTA$ | $ASR$ | $RA$ | $MTA$ | $ASR$ | $RA$ |
| Clean | 85.97 | 1.86 | 86.13 | 94.86 | 1.18 | 94.86 |
| No defense | 88.51 | 90.04 | 9.79 | 96.81 | 91.61 | 8.12 |
| *Beacon* | **92.94** | **0.52** | **92.77** | **98.37** | **0.13** | **98.37** |

Table 5: Robustness under dynamic attacks. BEACON consistently suppresses ASR and preserves RA across datasets.

## H    LIMITATIONS AND FUTURE WORK

While BEACON demonstrates strong robustness against backdoor attacks in cross-domain federated fine-tuning, it still has certain limitations. First, our evaluations are limited to image classification tasks, leaving the extension to multimodal federated fine-tuning scenarios and diverse backbone architectures (e.g., large vision–language models) as promising directions for future work. Second, BEACON may occasionally misclassify benign clients as suspicious, which could reduce the diversity of training updates and marginally impact model generalization. Future work will focus on refining the anomaly scoring mechanism to further reduce false positives while maintaining strong robustness.

## I    VISUALIZATION RESULTS OF ANOMALY SCORE

We further present visualization results of the scores computed by the TDOD and CHIF modules. In all cases, clients 3 and 6 are the truly malicious clients, and we highlight the indices identified by BEACON as malicious in red for comparison. As shown in Figs. 6–9 across different datasets and three representative attack variants, both TDOD and CHIF consistently assign higher anomaly scores to malicious clients, effectively separating them from benign ones. These visualizations provide intuitive evidence that BEACON reliably detects outliers and captures the subtle behaviors associated with backdoor manipulations.

## I.1 DOMAINNET

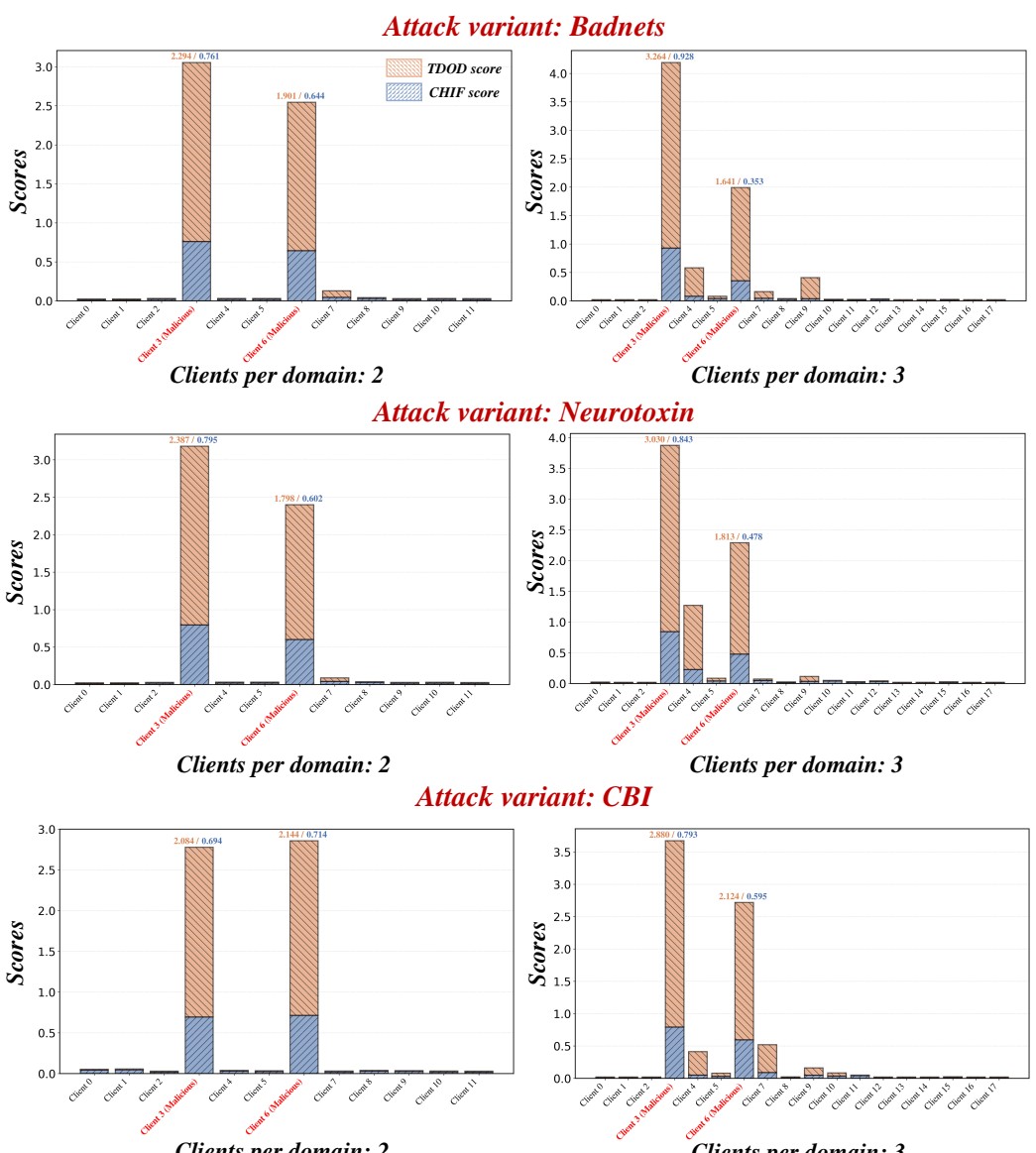

Figure 6: TDOD ($\Phi$) and CHIF ($\psi$) anomaly score visualizations on **DomainNet**. Clients 3 and 6 are the truly malicious clients. Indices identified by BEACON as malicious are highlighted in red.

## I.2 PACS

Figure 7: TDOD ($\Phi$) and CHIF ($\psi$) anomaly score visualizations on **PACS**. Clients 3 and 6 are the truly malicious clients. Indices identified by BEACON as malicious are highlighted in red.

## I.3 OFFICE-HOME

Figure 8: TDOD ($\Phi$) and CHIF ($\psi$) anomaly score visualizations on **Office-Home**. Clients 3 and 6 are the truly malicious clients. Indices identified by BEACON as malicious are highlighted in red.

## I.4 OFFICE-CALTECH-10

Figure 9: TDOD ($\Phi$) and CHIF ($\psi$) anomaly score visualizations on **Office-Caltech-10**. Clients 3 and 6 are the truly malicious clients. Indices identified by BEACON as malicious are highlighted in red.

