# OpenReview forum: "Beacon: Thwarting Backdoor Attacks in Cross-Domain Federated Fine-Tuning via Gradient Behavior Decoupling"
_ICLR.cc/2026/Conference — ICLR 2026 Conference Withdrawn Submission_

### Official Review · Reviewer_ek3D · 2025-10-22

**Soundness:** 2
**Presentation:** 2
**Contribution:** 2
**Rating:** 4
**Confidence:** 4

**Summary:**

This paper presents Beacon, a new framework for defending against backdoors in cross-domain federated fine-tuning. The authors argue that existing federated learning defences are ineffective due to domain shifts and parameter-efficient adaptation. Beacon introduces two core modules: TDOD decomposes client updates into task-consensus and domain-deviation components in order to isolate malicious signals, while CHIF detects targeted backdoor manipulations by analysing label-wise gradient inconsistencies in the classifier head. Extensive experiments on four datasets and three comparison backdoor attacks demonstrate that Beacon suppresses attack success while maintaining high main task accuracy, outperforming seven other baseline defences.

**Strengths:**

Experimental evaluation of existing FL backdoor defenses on CD-FFT clearly demonstrates their ineffectiveness.

Strong backdoor defense performance is achieved under the experimental settings.

The workflow of BEACON is clearly illustrated in Fig. 1, enabling readers to quickly understand the overall pipeline.

The security of CD-FFT, the main focus of this paper, represents an underexplored area in current research.

**Weaknesses:**

Federated fine-tuning is a widely used technique; however, its cross-domain variant is relatively uncommon. This paper provides limited background information on cross-domain federated fine-tuning (CD-FFT), making it difficult for readers to understand the contribution, particularly in Section 4.2.

The paper claims that CD-FFTs are vulnerable to backdoor attacks, but it does not discuss related work on existing backdoor attacks and defenses in the context of CD-FFT.

Compared with traditional federated learning (FL), the specific challenges of defending against backdoor attacks in cross-domain settings are not clearly explained.

Although the experiments show that most existing backdoor defenses perform poorly in CD-FFT, the underlying causes are not analyzed.

Several key parameters in TDOD and CHIF lack sensitivity analysis. The need to fine-tune all these parameters limits the practical applicability of this approach in real-world scenarios.

The proposed BEACON framework, particularly the CHIF module, may be vulnerable to specially designed attacks such as non-targeted or multi-target attacks.

While the paper reports the time cost of the proposed method under moderate conditions (few clients, medium-sized dataset), its scalability to large-scale FL settings remains unverified. Certain steps, such as the orthogonal decomposition in TDOD, may become computationally expensive as the model, dataset, or number of clients increases.

The leave-one-out strategy could be bypassed by coordinated, colluding adversaries.

Most comparative evaluations are based on defenses proposed before 2022, with only two recent backdoor detection methods included as baselines. This may weaken the comprehensiveness of the performance evaluation of BEACON.

**Questions:**

Current backdoor defenses often fail to effectively mitigate backdoor attacks in federated learning (FL) because most are designed for standard FL settings and do not generalize well to cross-domain or heterogeneous data distributions. The causes of poor performance vary, but common factors include reliance on global data similarity assumptions, sensitivity to non-iid data, and limited adaptability to diverse client behaviors.

The threshold \epsilon in TDOD is critical for filtering noisy gradient updates. Similarly, parameters \tau_m, \tau_s, \lambda_1, and \lambda_2 significantly influence the balance between detection sensitivity and stability. However, their sensitivity across different model architectures and datasets is not analyzed, and the paper lacks an ablation study to justify these parameter settings.

Since CHIF identifies label-specific anomalies in classifier head gradients, BEACON may be less effective against non-targeted or generalized backdoor attacks, which do not produce distinct gradient signatures. It is unclear how BEACON maintains robustness under such conditions, as its detection mechanism may not capture dispersed or non-class-specific deviations.

BEACON could also be vulnerable to colluding adversaries. If malicious clients coordinate their updates to imitate domain-specific variations, the leave-one-out consensus in TDOD could be deceived, causing the malicious updates to appear as benign and evade detection.

The computational efficiency and scalability of orthogonal component decomposition are not discussed. It remains uncertain whether BEACON can scale to large FL systems with more clients, larger datasets, or larger models.

Many state-of-the-art FL backdoor attacks and defenses proposed in recent years are not considered in this paper. It is unclear whether BEACON can still effectively mitigate these newer attacks and outperform contemporary defenses.

The paper also does not examine the impact of data distribution on BEACON’s performance. It is uncertain whether BEACON performs equally well under both iid and non-iid settings.

**Details Of Ethics Concerns:**

I don't see any ethical issues with this paper.

---

### Official Review · Reviewer_7Ech · 2025-10-31

**Soundness:** 2
**Presentation:** 3
**Contribution:** 2
**Rating:** 2
**Confidence:** 4

**Summary:**

The paper raises concerns about high susceptibility to backdoor attacks in cross-domain federated fine-tuning (CD-FFT), due to domain shift and rapid local adaptation. To solve this, it proposes BEACON, which decouples behavioral gradient into consensus and deviation components to detect malicious behaviors and capture intra-client boundary manipulations to enhance label-wise backdoor detection. The method is evaluated on ViT-Base/16 backbone and finetuned with visual prompt tuning across four datasets (DomainNet, PACS, OfficeHome, Office-Caltech10) and three backdoor variants (BadNets, Neurotoxin, CBI).

**Strengths:**

-	The paper focuses on cross-domain federated fine-tuning, which is an important but underexplored topic.
-	The main idea of separating benign task signals attack-induced deviations is conceptually elegant and well-motivated.
-	The experimental results are clearly summarized and the charts are well drawn.

**Weaknesses:**

-	At least, each component should be discussed to explain why it is expected to provide the preferred results.
-	No robustness bound or theoretical guarantee is given in the paper, which leaves uncertainty about stability under adaptive or stealthier attacks.
-	More analysis on scalability and complexity should be added. Both proposed modules in the paper may not scale well. For example, CHIF requires per-class gradient traversal, which scales with class count.
-	The paper assumes the attacker cannot adapt to the defense, which is unrealistic in federated settings.
-	Most of the baselines are pretty outdated, except for AlignIns. Furthermore, they are primarily designed for standard FL and not re-tuned for CD-FFT. This may overstate BEACON’s advantage.
-	The baselines are tested only on ViT-Base/16 backbone with visual prompt tuning. This raises concerns about BEACON's generalization across different architectures and fine-tuning strategies (E.g., LoRA, adapter tuning).
-	The experiment on varying numbers of clients is limited and not enough to show the effectiveness of the proposed method in a real-world large FL system.

**Questions:**

- How well can TDOD decouple gradient behaviors?
- Why TDOD decomposition effectively separates domain variance from malicious signals?
- What if the attacker knows BEACONS mechanisms (e.g., manipulating both consensus and deviation to evade detection)?

---

### Official Review · Reviewer_6fqi · 2025-10-31

**Soundness:** 3
**Presentation:** 3
**Contribution:** 3
**Rating:** 4
**Confidence:** 3

**Summary:**

This paper introduces BEACON, a novel backdoor defense framework designed for cross-domain model fine-tuning. BEACON computes a trust score composed of two key components: Task-Deviation Orthogonal Disentanglement, which decomposes each client’s update into consensus and deviation components, and Classification Head Inconsistency Forensics, which detects inconsistencies in clients’ decision boundaries. By combining these scores with a predefined threshold, BEACON adaptively assigns weights to clients, effectively mitigating the influence of malicious participants. Extensive experiments across multiple datasets and baseline methods demonstrate the effectiveness and robustness of BEACON.

**Strengths:**

- The paper is well written, clearly structured, and easy to follow.
- The proposed method BEACON is thoughtfully designed and clearly presented
- Authors conduct extensive experiments to validate the effectiveness of BEACON

**Weaknesses:**

Major:
- The experimental setup fixes malicious clients to specific indices (clients 3 and 6), which may limit the generality of the results
- The poisoning rate used in experiments (0.25) is relatively high and may not reflect realistic attack scenarios. This could make malicious clients easier to detect
- Lacks a sensitivity analysis of key hyperparameters $\lambda_1, \lambda_2, \tau_s, \tau_m$. Including such analysis would provide a more comprehensive understanding of BEACON’s robustness and stability

Minor:
- The overview figure is difficult to interpret due to the dense stripe patterns; simplifying the design would improve clarity
- Some key observations in Sections 4.1 and 4.3 would benefit from supporting experiments or illustrative examples to enhance reader intuition

**Questions:**

See weaknesses above

---

### Official Review · Reviewer_7dpN · 2025-11-01

**Soundness:** 3
**Presentation:** 3
**Contribution:** 3
**Rating:** 6
**Confidence:** 3

**Summary:**

The paper proposes BEACON, a novel server-side defense framework designed to identify malicious updates by decoupling gradient behaviors at a fine granularity. BEACON consists of two core modules which execute Task-Deviation Orthogonal Disentanglement (TDOD) to separate client updates into consensus and deviation parts for tasks and Classification Head Inconsistency Forensics (CHIF) to identify label tampering through gradient analysis. The research conducts testing of BEACON against three attack variants on four benchmarks which demonstrates its ability to defend against backdoor attacks at less than 2% success rate while maintaining task performance above 98% better than seven previous defense systems.

**Strengths:**

1.	It seems to be the first work to systematically address the critical vulnerability of cross-domain federated fine-tuning (CD-FFT) to backdoors, correctly identifying that existing defenses fail due to domain shift.
2.	The proposed BEACON framework uses a dual-module approach (TDOD and CHIF) to logically decouple benign domain variations from malicious gradient manipulations, which is an interesting idea.
3.	The research conducts multiple tests against seven security measures and three attack methods and four different datasets to demonstrate BEACON achieves attack success rates under 2% while maintaining task performance.

**Weaknesses:**

1.	BEACON introduces high computational and memory requirements because its TDOD module needs to perform additional calculations for consensus gradient storage and orthogonal basis computation which results in substantial weight increase compared to basic SOTA defenses like Krum or Trimmed-Mean.
2.	TDOD module fundamentally relies on the assumption that malicious gradient components are sufficiently orthogonal to the benign consensus; sophisticated attackers could design "stealthy" attacks that align their gradients more closely to the consensus to evade
3.	The evaluation lacks testing against sophisticated backdoor attacks which use adaptive methods to defeat the TDOD through orthogonal decomposition (e.g. making the malicious gradient component match the global update direction).
4.	The framework does not address other potential attack vectors, such as subtle poisoning of the initial pre-trained model or data-only poisoning attacks designed to make the malicious gradients mimic benign, domain-specific updates

**Questions:**

1.	How robust is BEACON against an adaptive attacker who designs a stealthy attack to minimize the orthogonal deviation component, thereby aligning the malicious gradient more closely with the benign global update direction?
2.	Can the authors provide an empirical comparison of the server-side latency and memory requirements of BEACON versus a computationally simpler SOTA defense like Trimmed-Mean or FLAME?
3.	How would BEACON perform against an advanced backdoor variant where the attacker only manipulates the feature extractor layers to embed the trigger, leaving the classifier head gradient largely uncorrupted?
4.	The anomaly thresholds are set using a percentile of assumed benign updates. How sensitive are the final results to the choice of this percentile, and how does BEACON maintain performance when the underlying data distribution experiences non-stationary domain drift over time, potentially shifting the benign update distribution?

---

### Note · Authors · 2025-11-13

I have read and agree with the venue's withdrawal policy on behalf of myself and my co-authors.